# Optimizing Platelet-Rich Plasma: Spin Time and Sample Source

**DOI:** 10.3390/bioengineering10111270

**Published:** 2023-10-31

**Authors:** Theodore E. Harrison, Jannice Bowler, Chin-I Cheng, Kenneth Dean Reeves

**Affiliations:** 1Independent Researcher, Victoria, BC V8L 5K1, Canada; 2Independent Researcher, Victoria, BC V9E 1J5, Canada; 3Department of Statistics, Actuarial and Data Science, Central Michigan University, Mt. Pleasant, MI 48859, USA; cheng3c@cmich.edu; 4Independent Researcher, Roeland Park, KS 66205, USA

**Keywords:** platelets, platelet-rich plasma, buffy coat

## Abstract

The buff-colored layer separating the plasma from red blood cells (RBCs) in centrifuged blood was named the “buffy coat” in the late 19th century. The division of platelets (PLTs) and leukocytes (WBCs) between the buffy coat, plasma, and RBC layers in centrifuged blood has not been described before. In this study, we centrifuged 8.5 mL anticoagulated blood samples at 1000× *g* for 1, 2, 3, 5, 10, and 20 min. We then divided each sample into ten layers and analyzed each layer for cellular composition and mean platelet volume (MPV). Our results show that even after 20 min of centrifugation, about 15% of platelets remain in the plasma layers and 65% in the RBC layers. We found that the platelet count achieved from aspiration of 1 mL volume was optimal, with aspiration beginning 1/2 mL below the buffy coat and extending 1/2 mL above the buffy coat rather than beginning at the buffy coat itself and aspirating only plasma. Using this method of aspiration, we found that the total platelet count means reached a maximum in the 1 mL around the buffy coat after only 5 min of centrifugation.

## 1. Introduction

The buffy coat is defined by Wikipedia as “the fraction of an anticoagulated blood sample that contains most of the white blood cells and platelets following centrifugation”. The term was first used in the late 19th century, and the characteristics of the buffy coat were elucidated by Powell in 1962 [1]. The technique of separation of blood components by density gradient centrifugation was developed in the 1960s and 1970s [2]. In the twentieth century, blood banks developed sophisticated methods for isolating the buffy coat as a source of platelets and white blood cells (WBCs) [3].

Kingsley first used the term “platelet-rich plasma” in 1954 [4], while Knighton et al. published the first case series in 1988 [5]. Platelet-rich plasma (PRP) was first defined by Marx in 2001 [6] as plasma with a platelet count of greater than 1,000,000 per microliter. While first promulgated in veterinary science and dentistry [7], its use spread rapidly to trauma surgery [8], orthopedics [9], plastic surgery [10], urology [11], ophthalmology [12], dermatology [13], regenerative medicine [14,15] and other specialties. Innovative new uses for PRP, such as in neurology [16,17], are being reported almost daily, and it appears that much of the progress in regenerative medicine is due to PRP.

The reason for the wide and increasing reach of PRP therapeutics is the discovery of the regenerative potential of platelets. Platelets release cytokines and growth factors when activated. The cytokines circulate in the blood and signal dormant stem cells to demarginate and enter the circulation. The stem cells home to the source of the cytokines where, under the influence of the growth factors and the local milieu, they duplicate and differentiate. In this way, new tissue is regenerated at the site of damage. The versatility of PRP is that this regenerative process can be instigated anywhere in the body where platelets can be injected and activated [18,19,20].

When prepared in the clinic, PRP is usually made by a simple centrifugation method, sometimes called the buffy coat method. There are numerous variations of this method, which relies on the density differential of cellular elements in the blood to allow for separation by centrifugation [21,22,23,24,25,26,27,28,29,30,31]. Numerous commercial machines have been developed to produce PRP [32,33,34,35], and because of the wide variability in the output of all these methods, several PRP classification systems have been proposed [36,37,38].

Every method of PRP production produces a different quality of PRP. Even beyond the basic differences in platelet concentration, RBC concentration, and WBC concentration, there are differences in cytokines, microvesicles, and probably hundreds of other factors [26,29,39,40]. Because there are so many variables, the definition of an optimum (or even a standard) PRP remains an elusive goal.

The most basic property of PRP is its platelet content. This depends on the platelet count of the blood from which the PRP is derived and the details of the preparation method. The controllable factors in PRP production by differential centrifugation are speed and time. Many different combinations have been used, but we could find no systematic comparison or analysis. If infinite time were available, then neither time nor speed would be a factor because all cellular elements would settle at the level of their density. Unfortunately, we have limited time in the clinic, and worse, platelets have a density range similar to that of WBCs. This makes it physically impossible to separate these two cell populations by density alone completely.

The basic technique is to centrifuge a sample of blood and extract the buffy coat along with a little plasma. However, in a previous study, we showed that the platelet yield from such single-spin methods does not exceed 72% [41]. This begs the question: where are the rest of the platelets, and why are they not in the buffy coat? Our hypothesis was that optimal aspiration methods for platelet-rich plasma would require aspiration below the buffy coat because the large, dense platelets in the blood below the buffy coat are unable to migrate to the buffy coat against the centrifugal force, and the opposing movement of migrating RBCs and are thus retained in the RBC layer. To confirm this hypothesis and how it might affect aspiration methods, we precisely measured platelet counts in layers above and below the buffy coat. We repeated these measurements by layer at varying times of centrifugation, as we also wanted to determine the optimum centrifugation time for PRP, which would result in the highest yield and best separation of PLTs from other blood components in a clinical setting.

## 2. Materials and Methods

After explanation of the study sixty sequential participants who were having blood drawn for PRP treatment of osteoarthritis signed informed consent and had one ethylenediaminetetraacetic acid (EDTA) tube (BD Vacutainer, K2 EDTA 7.2 mg, REF367861) and one acid citrate dextrose (ACD) tube (BD Vacutainer, ACD solution A, REF364606) of blood drawn for the study. A standard complete blood count (CBC) was performed by a hematology analyzer (MSLAB-7 Full-Auto Hematology Analyzer, Guangzhou Medsinglong Medical Equipment Co., Ltd., Guangzhou, China) on each EDTA tube as soon as possible. The ACD tubes were divided into six groups.

Each sample was centrifuged at 1000× *g*. After performing a literature review related to centrifugal force and spin time, we observed substantial variability in the literature related to the methods employed in stages of PRP processing [42] and a lack of consensus on optimal PRP preparation methods or for optimal concentration of blood components for various clinical indications [43]. We chose a centrifugal force of 1000× *g*, consistent with our previous study comparing six single-spin methods of platelet-rich plasma preparation. We also have observed that plastic syringes (used in some methods) tend to leak or break above that speed. In addition, 1000× *g* is approaching the maximum speed that most simple office centrifuges can produce, and we did not want to move into a g force range requiring a lab-level centrifuge. We used Rmid, the radius to the middle of the blood column, as the distance to compute centrifugal force since it is close to the target layer—the buffy coat—and to make our process more easily translatable to other centrifuges. For details on calculating Rmid, please see our previous publication on the mechanics of PRP centrifugation [44].

One group of ACD tubes was centrifuged for one minute, one group for two minutes, one group for three minutes, one for five minutes, one for ten minutes, and one for twenty minutes. After centrifugation, we measured the distance from the bottom of the tube to the top of the RBC layer and the top of the plasma layer with calipers. The first of these distances was divided by the second to obtain the ACD hematocrit. Then, each tube was marked into ten layers (starting from the buffy coat as the baseline), and each layer was pipetted sequentially from the top of the tube as an aliquot for analysis (Figure 1). The top and bottom layers (P5 and R5) were open-ended. I.e., their volume varied depending on the hematocrit and centrifugation time. The other layers were all 1 mL except the buffy coat layer and the layer below it (P1 and R1), which were only 0.5 mL because we were particularly interested in changes around the buffy coat.

Each aliquot was analyzed by a hematology analyzer in triplicate, with one measurement each from the top, middle, and bottom of the aliquot. The three measurements for WBC, RBC, hematocrit (Hct), platelets (PLT), and MPV for each aliquot were averaged, and the mean was entered into an Excel database.

### 2.1. Practical Guidance for Aspiration of 0.5 mL below and above the Plasma/RBC Junction

Aspiration of accurate volumes from test tubes or syringes requires precise measurements and markings. We recommend using a set of calipers accurate to within 0.01 mm. The inside diameter (ID) of the tube is measured with the calipers. The diameter is divided by two to obtain the radius and multiplied by π ($A=\pi r^{2}$) to attain the area of the circular cross-section of the tube. The desired volume is divided by the cross-sectional area to attain the length of the aspirated column necessary for the desired volume. Using the calipers, this length is measured and marked on the tube and aspirated between the marks. We started marking from the buffy coat and marked one 0.5 mL layer above and below the buffy coat and then four more 1 mL layers above and below those. Layers P5 and R5 were necessary for varying volumes because of the difference in hematocrit of each sample.

Aliquots were aspirated using the following technique: A 1 cc graduated syringe was attached to an 18 g × 3-inch hypodermic needle. The needle was used to pierce the yellow rubber stopper of a centrifuged sample tube; the syringe was removed briefly to equalize the pressure, and then the tip of the needle was advanced to the mark at the bottom of the layer to be aspirated so that the aperture of the needle was just above the mark. The needle was rotated so that the bevel faced inward, and the layer was gently aspirated until the meniscus in the tube reached the mark. The needle/syringe assembly was then withdrawn, and the actual volume was noted. The aliquot was then analyzed directly from the 1 cc syringe, taking one measurement from the top, middle, and bottom of the syringe and using the mean as the value for that sample.

Example: We have a tube with a 16 mm ID, and we want to aspirate 0.5 mL. The cross-sectional area is 201 square mm using the formula above. 0.5 mL/201 mm^2^ = 2.5 mm. Thus, we would mark off a section 2.5 mm long on the tube and aspirate that section to obtain the desired volume.

### 2.2. Statistical Analysis

Statistical analysis was conducted using R software version 4.3.0. The mixed-design ANOVA model considered for this study used layers as the within-subject variable and spin time as the between-subject variable. Platelet counts did not follow a normal distribution. The non-parametric mixed design ANOVA model by Brunner et al. [45] was employed to analyze platelet counts, and statistical inferences were based on relative effects. The relative effect for a group was estimated based on the arithmetic mean of the ranks in the group. A higher relative effect indicates a larger value. The relative effect of platelet counts among layers with 95% confidence intervals was provided to evaluate for significant differences between centrifuge times. The normality tests on residuals indicated that the mean platelet volumes were robust for the normality assumption. Therefore, the parametric mixed design ANOVA model was appropriate for the analysis of mean platelet volumes. All analytical results were considered significant when *p*-values were less than or equal to 0.05.

## 3. Results

### 3.1. Which Layers Have the Most Platelets?

Platelet counts over time by layer were non-parametrically distributed. For that reason, platelet counts are presented below in Table 1 as both means with standard deviations (SD) and as medians with interquartile ranges (IQR) for 1 mL plasma layers (P5-P2), 1 mL red cell layers (R2-R5), ½ mL plasma (P1), and ½ mL red cell layers (R1). Non-parametric statistical analysis revealed that platelet median concentration changes were significantly different according to layer (*p* < 0.001), time (*p* < 0.001), and layer by time (*p* < 0.001). Both mean and median platelet counts were higher in the P1 layer than in all other layers by five minutes. The P1 and R1 layer medians are higher than all other layers by 1 min of centrifugation.

Figure 2 shows the movement of platelets during the time of centrifugation. We clearly see the platelets in layers P2–P5 move to layer P1 by two minutes and increasingly through 5 min. Surprisingly, we see little movement in the platelets in layers R3–R5. A significant number of platelets remain trapped in layers R2–R5, and platelets in these layers did not appreciably move up to the buffy coat. Platelet numbers peak in the ½ mL plasma layer above the buffy coat (P1) at five minutes. With further centrifugation, many of these platelets are driven down into the ½ mL top red cell layer (R1), such that platelet numbers peak in the top red cell ½ ml layer at 20 min. Figure 3 and Figure 4 show the significant differences between platelet counts in the layers.

It is notable upon examination of this table that the platelet count in layer P1 rose from 10 to 20 min after dropping from 5 to 10 min and that the platelet count in Layer R5 nearly doubled from 10 to 20 min. A careful examination of 20 min data revealed an apparent data aberration in the P1 layer samples at the 20 min period. Absent those unexplained deviations, the P1 platelet count would likely have continued to drop at 20 min, but further data collection would be required for confirmation. A possible reason that platelet number increased in R5 at 20 min is that, at these long centrifugation times, the volume of R5 is more variable. See Figure 5, which shows the RBC layers become more compressed with longer centrifugation times. The R5 layer is open-ended.; i.e., the volume is whatever is left after R1–R4 has been aspirated. Thus, the number of platelets in the R5 layer depends more on the volume than the platelet concentration. Several samples had zero volume in R5 after 20 min (and were excluded from the calculations), so the dataset is also smaller.

### 3.2. Which 1 mL Layer Has the Highest Platelet Count, and Is It Significantly Higher Than Any Other Layer?

A stated goal of platelet concentration with a single-spin approach is to achieve a platelet count of 1000 (actually representing 1,000,000,000 platelets per mL). Commonly, this is accomplished by attempting to aspirate 1 mL of the most concentrated platelet layers.

The question of the best 1 mL layer to aspirate was analyzed by noting that the P1 and R1 layers had the highest concentrations, and each was only ½ mL. Those ½ mL layers were combined for analysis as a single layer, and, given the non-parametric nature of these data for this combination layer, it is presented here as both median (SD) and mean (IQR) values in Table 2.

The results here reveal that the highest platelet count for both mean and median, combining layers P1 and R1, is seen at 5 min, and this is despite the aberrantly high platelet counts in P1 at 20 min mentioned previously. The question is, do these differences between time periods reach statistical significance? Figure 3 is a relative effects table [45], which provides in one graph the relative effect and confidence intervals of median platelet counts across all time periods and all layers, with each layer representing 1 mL (including P1 + R1). For ease of interpretation, note that the layer P1 + R1 with each spin time, beginning by 3 min, has a relative effect that is significantly higher than any other 1 mL layer. For that reason, we can say with confidence that aspiration of the bottom ½ mL of plasma and the top ½ mL of cellular layers will result in the most favorable count among the 1 mL layers we directly measured.

However, common guidance (instruction) on how to aspirate the buffy coat in PRP preparation after a single-spin method is to begin at the buffy coat layer (the junction between plasma and red cell layers) and “vacuum” platelets off the buffy coat, aspirating few red cells; continuing aspiration until a 1 mL volume is achieved. This low cellular aspirate approach is equivalent to aspirating all of the ½ mL P1 layer and ½ mL of the 1 mL P2 layer. Table 3 presents mean (SD) and median (IQR) data for platelet counts that would be obtained by combining layers P1 and R1 layer and by combining P1 and ½ the P2 layer (“buffy coat” aspiration approach).

The differences between the P1 + R1 aspiration and the “buffy coat” aspiration approach appear to diverge at 5 min. However, is this difference in result from these two approaches statistically significant? Figure 4 is the relative effects figure comparing the relative effects of each aspiration approach across time intervals. When these two approaches are compared according to relative effects, it is notable that by 5 min, the overlap between confidence intervals is minimal, and the P1 + R1 approach is clearly favored by 10 min.

### 3.3. What Is the Optimum Centrifugation Time?

It appears that both medians and means are similar between groups when aspirating the P1 + R1 layers, consistent with the comparison of median platelet count results in Table 3. The pattern of similarity is particularly notable, comparing 5 min and 20 min values. Given the degree of similarity, it is likely that a very large data set would be required to show a difference in platelet concentration medians between five and twenty minutes of spin time. Our data suggest that there is no advantage of a longer centrifugation time than 5 min. The practicality of limiting centrifugation to five minutes is considerable, given a superior time efficiency, but this is dependent on visualization of the junction between plasma and red cells for practical aspiration. This appears to be adequate by five minutes in these glass test tubes, according to empiric observation.

### 3.4. Effects on Centrifugation Time on Hematocrit

Figure 5 shows a graph of the ratio of ACD hematocrit to whole blood (EDTA tube) hematocrit versus time (ACD hematocrit and EDTA hematocrit differ significantly because the blood in the ACD tube is diluted with anticoagulant.). This shows that even at twenty minutes of centrifugation, RBC compaction has still not found a stable level. However, by about five minutes, the vast majority of compaction has occurred.

### 3.5. Effects of Platelet Volume on Platelet Distribution

MPV distribution was parametric, allowing for the use of standard ANOVA analysis methods. With all time periods included for analysis, the MPV differed significantly between layers (*p* < 0.001 and over times of centrifugation (*p* < 0.001). Table 4 presents the estimated marginal means of MPV over 5, 10, and 20 min according to layers, along with their confidence intervals.

Although Figure 6 shows what appears to be a definite trend toward a higher platelet volume in the P1 + R1 and R2 layers, the confidence intervals in Table 4 overlap, suggesting that our study was too small to show a significant difference in platelet sizes between layers.

## 4. Discussion

In this pragmatic aspiration method study, we have demonstrated that the most concentrated platelet concentration obtainable after a single-spin method will be achieved by aspirating the top ½ mL of the cellular layer and the bottom ½ mL of the plasma layer in a centrifuged blood sample of about 7 mL. We have compared efficacy to the standard buffy coat aspiration method, with results indicating the superiority of the ½ mL above and ½ mL below aspiration approach. The most effective centrifugation time appears to be 5–20 min, with results so close that they suggest no clear advantage to a centrifugation time beyond 5 min. This is corroborated by the rate at which RBCs compact, which begins to flatten at five minutes.

The common perception is that centrifugation separates the cellular components and blood plasma into different layers based on density. Blood banks have devised sophisticated machines and procedures to take advantage of this property. These techniques, however, are beyond the capabilities of most clinics.

This study shows that at commonly used clinical centrifugation speeds and times, most PLTs and WBCs remain outside the buffy coat. However, their behavior depends on their location at the start of centrifugation. Platelets that start above the buffy coat tend to migrate rapidly down the tube to the buffy coat, although a few remain suspended. Platelets that start below the buffy coat have a much more difficult time. They must be pushed up to the buffy coat against centrifugal force by the denser RBCs. This is a slower process, and in the end, more are left behind. While some PLTs and WBCs migrate into the buffy coat, the majority remain behind in the RBC and plasma, although both PLTs and WBCs tend to migrate to the top of the RBC layer. This area appears red (and therefore is not technically part of the buffy coat) because it still has a hematocrit of 22–56% (depending on centrifugation time) regardless of the high concentration of PLTs and WBCs.

As expected, about 35% of PLTs remained trapped in the lower RBC layers after twenty minutes of centrifugation. However, surprisingly, 15% of platelets also remained in the upper plasma after twenty minutes. It is possible that centrifugation at higher speeds and/or longer times would produce better yields, but this would require more expensive equipment and longer wait times in the clinic. It seems likely that methods other than single-spin centrifugation will be needed to enhance successful separation of platelets from other blood components [46].

The MPV analysis suggests that large, dense PLTs migrate rapidly from the upper plasma to the top of the RBC layer. However, it disproves our hypothesis that these PLTs are preferentially slowed in migration from the layers below the buffy coat. It seems that migration of all sizes of PLTs is substantially slowed by centrifugal force and the opposing motion of migrating RBCs.

Miron et al. [47] evaluated the effect of various g forces for various spin times. There were several key differences in their approach. First, Miron et al. evaluated by aspirating 1 mL volumes beginning at the top of the tube. This ignores differences in hematocrit, which cause the buffy coat to be at a variety of levels. In contrast, we measured from the buffy coat. Our focus was on the migration of blood components with respect to the buffy coat. Given that attention in PRP preparation is focused on the buffy coat, our hope was that our findings would be more pertinent to the clinician. Second, we evaluated the effect of centrifugation beginning at a shorter centrifugation time (1 min) and extending longer (20 min) to provide a more extended observation of the migration of blood components in response to a commonly utilized g force (1000× *g*). Third, we evaluated our results statistically to search for significant differences between layers, with an emphasis on platelet counts.

In the literature, a focus of interest has been the difference in platelet counts and other cellular components between single-spin and double-spin methods of PRP production. Magalon et al. [48] evaluated a number of devices that achieved a higher platelet capture rate than typically seen with single-spin methods but admitted that the limitation of their scoring system for PRP preparation methods is that the impact of platelet dose and purity remains unknown. In addition, no device is capable of recovering more than 90% of the platelets [48], and cost differentials that bear on cost efficacy were not included in scoring methods. Fadadu et al. [38]. echoed Magalon et al.’s conclusions in their 2018 review, concluding that the large heterogeneity between separation systems (single-or-double-spin) must be resolved for conclusions about relative clinical efficacy. Oudelaar et al. [34] pointed out that the choice as to the most appropriate system for PRP preparation is dependent on its intended clinical application, so that studies showing merely a difference between single-and-double-spin methods cannot, by themselves, inform clinical choices [49].

Observations in the current study and recent publications on single-spin methods appear useful in moving toward consistent PRP preparation methods for single-spin and potentially double-spin approaches. In this publication, we observed significant differences in yield that result from the aspiration method, specifically with respect to the region of the buffy coat. This is likely generalizable to any method that relies on buffy coat separation. In another recent publication, we described an app to help with PRP preparation [44], emphasizing that applying published g forces from this or any other article to day-by-day clinical use depends on measures specific to each brand of centrifuge (rotor position), depth of tubes in use, and hemoglobin.

Similar to other fields of medicine, in addition to obtaining condition-specific guidance on optimal parameters for PRP components such as platelet count, WBC count, RBC count, and cytokines), it will be important to consider the relative cost efficacy of manual single-or-double-spin methods versus proprietary single-or-double-spin methods.

## 5. Conclusions

This study confirmed that migration of platelets to the buffy coat from both plasma and RBC layers during centrifugation of blood at 1000× *g* for up to twenty minutes is incomplete, with no obvious difference in rate according to platelet size. Platelets above the buffy coat level migrate rapidly down to the buffy coat by five minutes of centrifugation. Further, centrifugation is less efficient at inducing the migration of platelets below the buffy into the buffy coat but increases the total number of platelets in the layers below the buffy coat interface. Even after 20 min of centrifugation, platelets remaining in RBC layers amount to approximately 30% of the total platelets. Thus, it is unlikely that any single-spin PRP method will have a yield greater than about 70%. More sophisticated techniques must be developed to produce more efficient yields and less RBC/WBC-contaminated PRP.

## Figures and Tables

**Figure 1 bioengineering-10-01270-f001:**
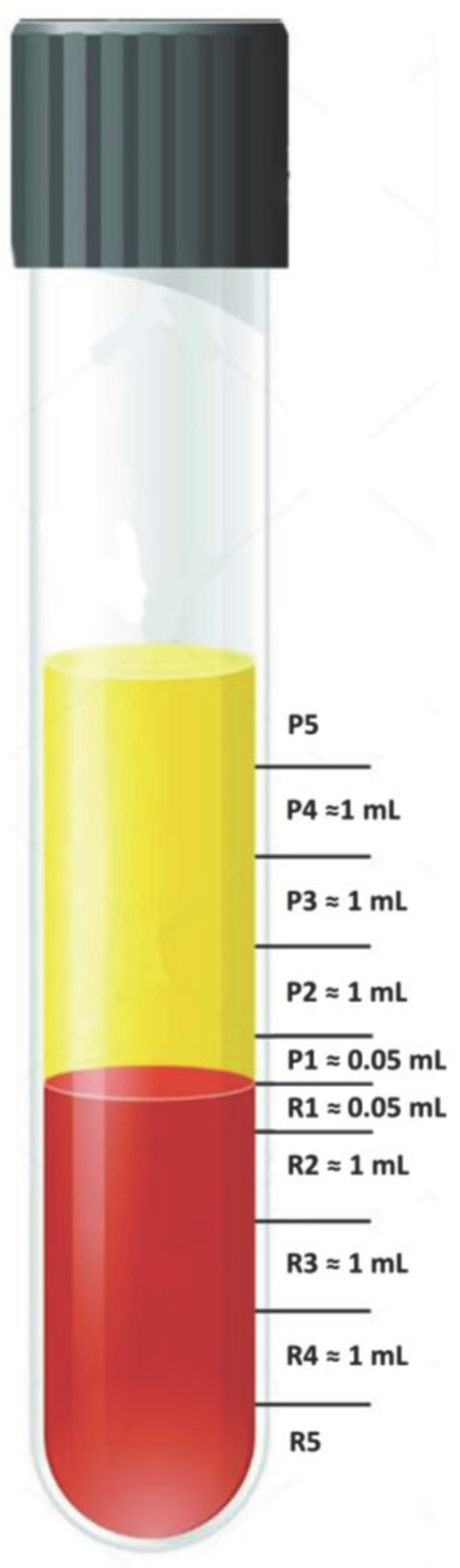
After centrifugation, each tube was divided into ten aliquots—five in the plasma layer (labeled P5–P1) and five in the RBC layer (labeled R1–R5).

**Figure 2 bioengineering-10-01270-f002:**
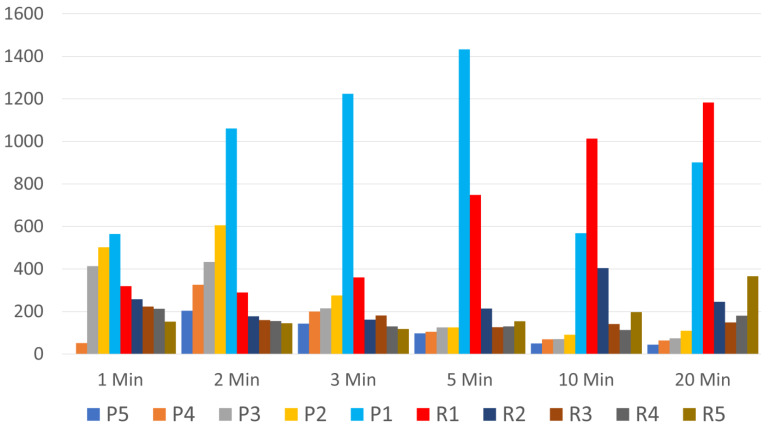
Number of platelets in each layer by time.

**Figure 3 bioengineering-10-01270-f003:**
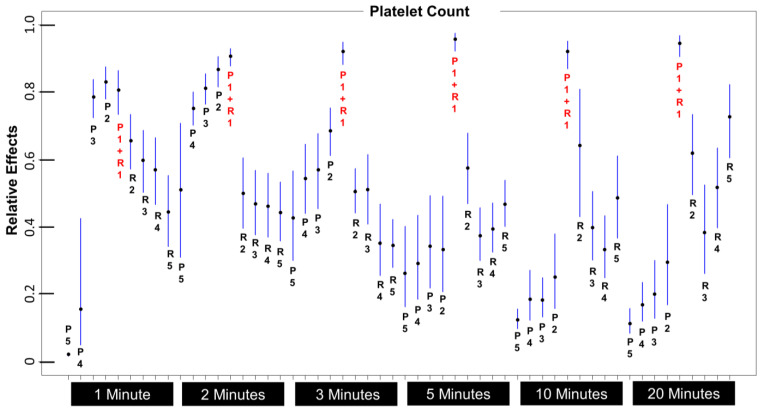
Median platelet counts across time periods and layers. Note that P1 + R1 is significantly higher from 3 min onwards.

**Figure 4 bioengineering-10-01270-f004:**
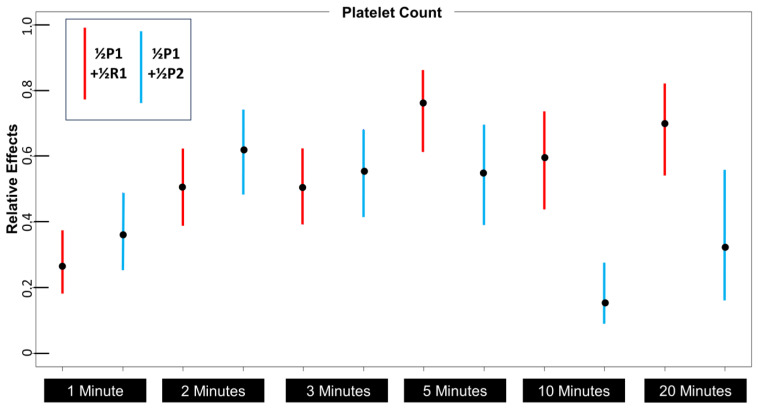
Platelet count relative effects comparison between layers P1 + 1/2P2 and P1 + R1 over time. Divergence occurs between five and ten minutes.

**Figure 5 bioengineering-10-01270-f005:**
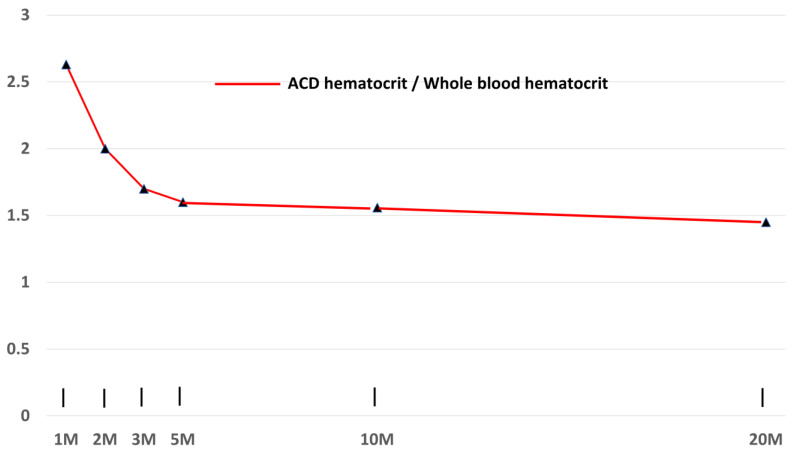
ACD:WB (whole blood) hematocrit ratio versus time. Maximum RBC compression is almost complete by five minutes of centrifugation.

**Figure 6 bioengineering-10-01270-f006:**
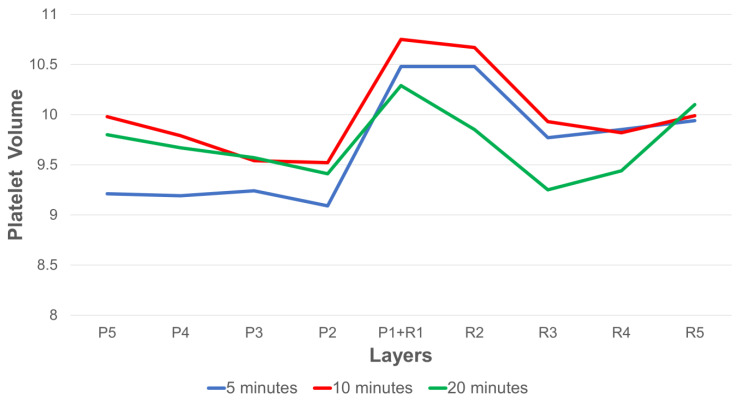
MPV by layer by time.

**Table 1 bioengineering-10-01270-t001:** Mean and median platelet counts by layer by centrifugation time.

Layer	Means (SDs) According to Time in Minutes					
	1	2	3	5	10	20
P5PLT	0 (0)	204 (49)	143 (25)	97 (22)	50 (6)	44 (10)
P4PLT	52 (35)	326 (32)	200 (34)	105 (23)	69 (13)	64 (12)
P3PLT	414 (63)	433 (54)	215 (34)	125 (29)	70 (11)	74 (15)
P2PLT	502 (71)	606 (103)	276 (38)	125 (30)	91 (19)	110 (32)
P1PLT	565 (73)	1061 (107)	1224 (163)	1433 (239)	568 (86)	901 (250)
R1Cell	319 (45)	290 (41)	360 (78)	748 (176)	1013 (190)	1183 (143)
R2 Cell	258 (34)	178 (24)	162 (11)	214 (30)	404 (119)	246 (38)
R3 Cell	223 (31)	160 (17)	181 (29)	126 (10)	141 (18)	149 (35)
R4 Cell	213 (34)	155 (16)	130 (22)	130 (8)	113 (12)	180 (25)
R5 Cell	152 (18)	145 (12)	118 (8)	154 (15)	197 (46)	366 (66)
**Layer**	**Medians (IQRs) according to time in minutes**					
	**1**	**2**	**3**	**5**	**10**	**20**
P5PLT	0 (0)	232 (353)	152 (101)	81 (100)	45 (25)	40 (57)
P4PLT	0 (55)	316 (142)	158 (119)	77 (96)	55 (64)	61 (75)
P3PLT	357 (357)	410 (231)	187 (150)	109 (148)	51 (76)	57 (59)
P2PLT	407 (404)	501 (424)	242 (76)	110 (129)	79 (89)	71 (114)
P1PLT	517 (309)	1099 (653)	1287 (923)	1147 (1224)	471 (309)	526 (1315)
R1Cell	313 (212)	272 (246)	281 (363)	504 (748)	779 (1036)	1298 (606)
R2 Cell	226 (211)	157 (139)	154 (38)	192 (159)	273 (459)	235 (125)
R3 Cell	188 (192)	146 (104)	162 (72)	127 (47)	116 (71)	118 (86)
R4 Cell	175 (165)	142 (84)	110 (45)	130 (28)	114 (67)	157 (121)
P5PLT	149 (95)	154 (67)	121 (41)	139 (26)	135 (103)	366 (299)
P4PLT	0 (0)	232 (353)	152 (101)	81 (100)	45 (25)	40 (57)

**Table 2 bioengineering-10-01270-t002:** Platelet counts by centrifugation time in the combined 1 mL layer consist of a combination of the ½ mL P1 and the ½ mL R1 layers. Note that the median values are different than the mere addition of medians of P1 and R1, which is to be expected.

Layer	Means (SDs) According to Time in Minutes					
	1	2	3	5	10	20
**1/2P1 + 1/2R1 PLT**	442 (58)	676 (65)	792 (92)	1090 (122)	790 (87)	1042 (153)
	**Medians (IQRs) according to time in minutes**					
**1/2P1 + 1/2R1 PLT**	418 (269)	682 (434)	774 (448)	1063 (749)	786 (446)	888 (799)

**Table 3 bioengineering-10-01270-t003:** Platelet counts comparison overtime for P1 + R1 (each ½ mL) and P1 + ½ of the 1 mL P2 layer.

Layer	Means (SDs) According to Time in Minutes					
	1	2	3	5	10	20
**1/2P1 + 1/2R1 PLT**	442 (58)	676 (65)	792 (92)	1090 (122)	790 (87)	1042 (153)
**1/2P1 + 1/2P2 PLT**	533 (69)	834 (98)	750 (92)	779 (120)	330 (49)	506 (126)
**Layer**	**Medians (IQRs) according to time in minutes**					
**1/2P1 + 1/2R1 PLT**	418 (269)	682 (434)	774 (448)	1063 (749)	786 (446)	888 (799)
**1/2P1 + 1/2P2 PLT**	513 (366)	817 (594)	759 (447)	657 (682)	298 (151)	307 (745)

**Table 4 bioengineering-10-01270-t004:** MPV by layer by centrifugation time.

Time	Layer	Mean	Std. Error	95% C.I. Lower Bound	95% C.I. Upper Bound
5	P5	9.210	0.190	8.829	9.591
	P4	9.190	0.228	8.733	9.647
	P3	9.240	0.227	8.784	9.696
	P2	9.090	0.219	8.650	9.530
	P1 + R1	10.480	0.224	10.032	10.928
	R2	10.475	0.277	9.919	11.031
	R3	9.770	0.202	9.365	10.175
	8	9.850	0.220	9.409	10.291
	9	9.940	0.189	9.561	10.319
10	P5	9.980	0.190	9.599	10.361
	P4	9.790	0.228	9.333	10.247
	P3	9.540	0.227	9.084	9.996
	P2	9.520	0.219	9.080	9.960
	P1 + R1	10.750	0.224	10.302	11.198
	R2	10.670	0.277	10.114	11.226
	R3	9.930	0.202	9.525	10.335
	8	9.820	0.220	9.379	10.261
	9	9.990	0.189	9.611	10.369
20	P5	9.800	0.190	9.419	10.181
	P4	9.670	0.228	9.213	10.127
	P3	9.570	0.227	9.114	10.026
	P2	9.410	0.219	8.970	9.850
	P1 + R1	10.290	0.224	9.842	10.738
	R2	9.850	0.277	9.294	10.406
	R3	9.250	0.202	8.845	9.655
	8	9.440	0.220	8.999	9.881
	9	10.100	0.189	9.721	10.479

## Data Availability

Not available.

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
