# Peer review of "Optimizing Platelet-Rich Plasma: Spin Time and Sample Source"

_bioengineering, 2023, doi:10.3390/bioengineering10111270_

Round 1

Reviewer 1 Report

Comments and Suggestions for Authors

The topic is intriguing, and the manuscript is well-written with good flow. For final publication, I highly recommend making the following minor adjustments:

Fig 2, 5 and 6 are trying to visualize the changes over time. But the x-axis is not linear, which can be misleading especially when talking about the critical time point. It is recommended to re-do the x-axis intervals. 

Consider adding a statement in the introduction about the most commonly used centrifuge speed and time in the cited publications.

The study's conclusion primarily relies on platelet count in different layers. It would be beneficial to include discussions about activity and cytokine contents.

Please review and correct the caption for Figure 5, as it appears to contain errors.

Author Response

Dear Reviewer,

Thank you very much for reading and commenting on our paper. We truly appreciate your suggestions. We had the following specific responses:

Comment 1: Figures 2, 5 and 6 are trying to visualize the changes over time. But the x-axis is not linear, which can be misleading especially when talking about the critical time point. It is recommended to re-do the x-axis intervals. 

Response 1: Figure 2 has been deleted as unnecessary given the mean and median table 1 and the new figure 2 which shows patterns of mean platelet counts over all time intervals. Figure 5 has been deleted as well as redundant with table 2 which compares mean and median platelet counts over all time intervals. B: Figure 5 has an X axis which represents layers rather than time. Due to that it could not be modified to a linear axis. Figure 6 (Renumbered figure 5) axis was converted to linear.

Comment 2: Consider adding a statement in the introduction about the most commonly used centrifuge speed and time in the cited publications.

Response 2: We added the following statement (cited) in the introduction about centrifuge speed and times:

Upon literature review related to centrifugal force and spin time, we observed substantial variability in the literature related to the methods employed in stages of PRP processing, and a lack of consensus on optimal PRP preparation methods or for optimal concentration of blood components for various clinical indications. We chose a centrifugal force of 1000g, consistent with our previous study comparing six single-spin methods of platelet rich plasma preparation. We also have observed that plastic syringes (used in some methods) tend to leak or break above that speed. In addition, 1000g is approaching the maximum speed that most simple office centrifuges can produce, and we did not want to get into a g force range requiring a lab level centrifuge.

Comment 3: The study's conclusion primarily relies on platelet count in different layers. It would be beneficial to include discussions about activity and cytokine contents.

Response 3: We would have liked to include measurements of cytokine activity and content, but, unfortunately, it was beyond our capabilities.

Comment 4: Please review and correct the caption for Figure 5, as it appears to contain errors.

Response 4: Figure 5 was deleted as redundant. 

Thanks again for your time and help.

Regards,

Ted Harrison

Reviewer 2 Report

Comments and Suggestions for Authors

Dear authors, nice topic.

1. Marx first to allocate PRP 1 bio platelet /ml as clinical relevant. First mentioned in literature term PRP was by Kingsley in 194. First PRP clinical protocol by Knighton in 1980's

2. Primary focus is on single spin, issues with single spin are well know from capture rate perspective. PRP is also made by double spin more effective devices with higher capture rates (indicated by Magalon and Fadadu).

I notice many "older references" suggest to update reference list with later than 1990's.

Your paper resembles very much the paper of Miron et al. 2020 20:310, in BMC Oral Health.

Would this have been an adequate reference for your method?

3. Line 69 statement is misleading not exceed more than 70%. Omajority single spin device platelet capture is no more than 40%, same references as in 2 and others

4. double spin devices overcome many of issues with single spin as MPV and PDW are all subject to a second spin; ie, higher layer platelets, discarded with single spin devices, continue to be concentrated. More control of platelet levels.

5. line 107: describe how aliquotes were taken: device used and sample direction.

6. sample distances are very narrow in small test tube. How can you be certain that there is no mixing from other layers? Clarify in methods please.

7. line 158: i agree on the goal of available platelets of 1 bio/ml. However, what is the relevance with this device, as 1 ml of PRP for therapeutics is  a limited field of application. (your conclusion is similar to the paper from Miron 2020 regarding 24 protocols)

8. Can you discuss/reason why the more dense platelets are in higher plasma layers after 10 and 20 min would one not expect that density locates them closer to R1. 

After 10 and 20 min the P5 MPV has increased, denser platelets move thus upwards from buffy coat to top layer P5?

9. Inline 265 you state that dense platelets migrate rapidly from upper down to RBC layer, which has also been seen by Ozer et al. In their paper less dense platelets are in upper of test tube, further down to BC has highest MPV. Your data show opposite

Clarify please why P2 has less dense platelets compared to P5, with increase over time.

10. Your claim in line 262 needs references!

11. Your conclusion should be more explicit, as this only affects the test tube device used in this method and might be different for other devices.

12. The impact, and current discussions, for potentially collecting cells/plasma from P1 to R1 has clinical implications for the bioformulation of the PRP; comment on the density of other cells in this regard as the PRP biology changes with their presence, as compared to a red-ish free PRP specimen.

Comments on the Quality of English Language

no comments here

Author Response

Dear Reviewer,

Thank you very much for your excellent critique of our paper. We truly appreciate the time you put into it and your helpful suggestions. Please find detailed responses as follows.

Comment 5: Marx first to allocate PRP 1 bio platelet /ml as clinically relevant. First mentioned in literature term PRP was by Kingsley in 194. First PRP clinical protocol by Knighton in 1980's

Response 5: Thank you for recommending that we point out the historical antecedents of PRP. We added the following content to the introduction:

Kingsley first used the term “platelet-rich plasma” in 1954[4], while Knighton et al published the first case series in 1988[5]. Platelet-rich plasma (PRP) was first defined by Marx in 2001 [6] as plasma with a platelet count of greater than 1,000,000 per microliter. While first promulgated in dentistry [7], its use spread rapidly to trauma surgery [8], orthopedics [9], plastic surgery [10], urology [11], ophthalmology [12], dermatology [13], regenerative medicine [14, 15] and other specialties. Innovative new uses for PRP, such as in neurology [16, 17], are being reported almost daily and it appears that much of the progress in regenerative medicine is due to PRP.

Comment 6: Primary focus is on single spin, issues with single spin are well known from a capture rate perspective. PRP is also made by double spin by more effective devices with higher capture rates (indicated by Magalon and Fadadu).

Response 6: Thank you for your prompting to add a component to the discussion related to the single versus double spin issue.   We added the following content to the discussion.

In the literature a focus of interest has been on the difference in platelet counts and other cellular components between single spin and double spin methods of PRP production. Magalon et al (2016) Magalon J, Chateau AL, Bertrand B, Louis ML, Silvestre A, Giraudo L, Veran J, Sabatier F. DEPA classification: a proposal for standardising PRP use and a retrospective application of available devices. BMJ Open Sport Exerc Med. 2016 Feb 4;2(1):e000060. doi: 10.1136/bmjsem-2015-000060. PMID: 27900152; PMCID: PMC5117023 evaluated a number of devices that achieved a higher platelet capture rate than typically seen with single spin methods but admitted that the limitation of their scoring system for PRP preparation methods is that the impact of platelet dose and purity remains unknown. In addition to limited information about clinical efficacy comparison studies are quite limited, no device is capable of recovering more than 90% of the platelets, (Magalon 2016) and cost differentials that bear on cost efficacy were not included in scoring methods. Fadadu Fadadu PP, Mazzola AJ, Hunter CW, Davis TT. Review of concentration yields in commercially available platelet-rich plasma (PRP) systems: a call for PRP standardization. Reg Anesth Pain Med. 2019 Apr 16:rapm-2018-100356. doi: 10.1136/rapm-2018-100356. Epub ahead of print. PMID: 30992411. echoed Magalon et al’s conclusions in their 2018 review concluding that the large heterogeneity between systems of separate must be resolved for conclusions about relative clinical efficacy. Oudelaar et al. pointed out that the choice as the most appropriate system for PRP preparation Oudelaar BW, Peerbooms JC, Huis In 't Veld R, Vochteloo AJH. Concentrations of Blood Components in Commercial Platelet-Rich Plasma Separation Systems: A Review of the Literature. Am J Sports Med. 2019 Feb;47(2):479-487. doi: 10.1177/0363546517746112. Epub 2018 Jan 16. PMID: 29337592.   is dependent on its intended use, the specific clinical field of application, and this information is yet to be determined, so that studies showing merely a difference between single and double spin methods cannot, by themselves, inform clinical choices. Saqlain N, Mazher N, Fateen T, Siddique A. Comparison of single and double centrifugation methods for preparation of Platelet-Rich Plasma (PRP). Pak J Med Sci. 2023 May-Jun;39(3):634-637. doi: 10.12669/pjms.39.3.7264. PMID: 37250535; PMCID: PMC10214802

Observations in the current study and recent publications on single spin methods appear to be useful in moving toward consistent PRP preparation methods for single spin and potentially double spin approaches. In this publication we observed significant differences in yield that result from the aspiration method, specifically with respect to the region of the buffy coat. This is likely to generalize to any manual method that requires aspiration. In another recent publication we described an app to help with PRP preparation, emphasizing that to apply published g forces from this or any other article to day-by-day clinical use depends on measures specific to each brand of centrifuge (rotor position), depth of tubes in use, and hemoglobin.

Comment 7: I notice many "older references" suggest to update reference list with later than 1990's.

Response 7: References have been updated in multiple areas.

Comment 8: Your paper resembles very much the paper of Miron et al. 2020 20:310, in BMC Oral Health. Would this have been an adequate reference for your method?

Response 8: Miron et al. (2020) is an excellent reference. We are grateful for its mention and we have included a discussion of the contrasts between our article and Miron et al as follows:

Miron et al. evaluated the effect of various g forces for various spin time, with a number of key differences from our approach. First Miron et al. evaluated by aspirating 1 ml volumes from the top of the tube. This ignores differences in hematocrit which cause the buffy coat to be at a variety of levels. In contrast, we measured from the buffy coat. Our focus was on migration of blood components with respect to the buffy coat. Given that attention in PRP preparation is focused on the buffy coat, we were expecting that our findings would provide more pertinent recommendations for manual aspiration. Second, we evaluated a wider range of centrifugation times, beginning at a shorter centrifugation time (1 minute), and extending longer (20 minutes), to provide a more extended view over time of migration of blood components in response to a commonly utilized g force clinically (1000g). Third we evaluated our results statistically to search for significant differences between layers, with an emphasis on platelet counts, and applied that information to recommendations on methods of aspiration.

Comment 9: Line 69 statement is misleading That the platelet yield from single-spin methods does not exceed 70%. Majority of single spin device platelet capture is no more than 40%, same references as in 2 and others.

Response 9: As we published in Harrison et al. in 2020, comparing 6 single spin methods, the average yield for a single spin method varied from 53±18 to 72±13%. Given that the yield of all single spin methods evaluated exceeded 50%, and 3 met or exceeded 60% (64±18, 65±17, and 72±12%), our statement was not misleading. No change in the text appears to be indicated.

Comment 10: Double spin devices overcome many of issues with single spin as MPV and PDW are all subject to a second spin, i.e. higher layer platelets, discarded with single spin devices, continue to be concentrated. More control of platelet levels.

Response 10: The purpose of this study was not to test different PRP methods or to promote a particular method, but to ascertain the behavior of platelets during the centrifugation process, pertinent to both single and double spin methods. To clarify our purpose: we wanted to watch the behavior of platelets over time by centrifuging at sequentially longer times and analyzing where the platelets were at each time point.

We added a section in the discussion to address double spin versus single spine as per response 6 and explained our focus in the discussion as per response 8.

Comment 11: Line 107: describe how aliquots were taken: device used and sample direction.

Response 11: We added the following text to the Methods section:

Aliquots were aspirated using the following technique: A 1cc graduated syringe was attached to an 18g x 3-inch hypodermic needle. The needle was used to pierce the yellow rubber stopper of a centrifuged sample tube; the syringe was removed briefly to equalize the pressure, and then the tip of the needle was advanced to the mark at the bottom of the layer to be aspirated so that the aperture of the needle was just above the mark. The needle was rotated so that the bevel faced inward, and the layer was gently aspirated until the meniscus in the tube reached the mark. The needle/syringe assembly was then withdrawn and the actual volume noted. The aliquot was then analyzed directly from the 1cc syringe, taking one measurement from the top, middle, and bottom of the syringe and using the mean as the value for that sample.

Comment 12: Sample distances are very narrow in small test tube. How can you be certain that there is no mixing from other layers? Clarify in methods please.

Response 12: Please see Response 11.

Comment 13: Line 158: I agree on the goal of available platelets of 1 billion/ml. However, what is the relevance with this device, as 1 ml of PRP for therapeutics is a limited field of application. (your conclusion is similar to the paper from Miron 2020 regarding 24 protocols)

Response 13: 1 mL of 4x concentrated PRP does not appear to be a limited field of application, as that 1 mL is obtained from 7 mL of plasma. A typical 12 bay inexpensive centrifuge from 84 mL of plasma can thus provide 12 ml of a billion/ml PRP with ACD tubes costing 24$, and a simple blood draw kit. In contrast, 50 mL of serum, using a $2800 Emycte and a proprietary 360$ kit and tubes, an Eppendorf centrifuge can produce 9-10 mL of 4-4.5X concentration of PRP, which would translate to 12 mL of adequate PRP per 60 ml plasma. Thus the capture rate may approximate 80%. Cost effectiveness is questionable to obtain the 10-15% additional capture rate. In addition, the higher concentration platelet options have not been adequately researched for cost effectiveness either. Given this, we added the following comment at the end of the discussion section.

“Similar to other fields of medicine, in addition to obtaining condition-specific guidance on optimal parameters for PRP components such as platelet count, WBC count, RBC count and cytokines), it will be important to consider the relative cost efficacy of manual single or double spin methods versus proprietary single or double spin methods.”           

Comment 14: Can you discuss/reason why the more dense platelets are in higher plasma layers after 10 and 20 min would one not expect that density locates them closer to R1.

Response 14: Thank you.  We agree that there is no logical explanation why the mdensee platelets are in higher plasma layers after 10 and 20 min. All we can do is attribute this to sampling variability which did not reach statistical significance.          

Comment 15: After 10 and 20 min the P5 MPV has increased, denser platelets move thus upwards from buffy coat to top layer P5?

Response 15: Thank you. Again, these are not statistically significant differences so we can only attribute them to sampling variability.

Comment 16: In line 265 you state that dense platelets migrate rapidly from upper down to RBC layer, which has also been seen by Ozer et al. In their paper less dense platelets are in upper of test tube, further down to BC has highest MPV. Your data show opposite.

Response 16: Ozer et al. (2019) did not aspirate by layers, but merely aspirated 1 mL from the lowest plasma layer above the buffy coat and specifically did not enter the buffy coat so this was a P1 layer without the buffy coat. Thus they did no investigation of layering effects on MPV. In addition, they only had 1 sample for each spin time and centrifuge speed, so there would be no potential for statistical significance or to come to a significant conclusion across participants as there was effectively only 1 participant.

Comment 17: Clarify please why P2 has less dense platelets compared to P5, with increase over time.

Response 17: Thank you. This is another instance where we think we had insufficient data. The differences are not statistically significant and our sample size was relatively small

Comment 18: Your claim in line 262 needs references!

Response 18: This referenced our own unpublished attempts at finding a two-spin method that matched the yields of our single-spin experiments. Since we have not published that data we deleted that sentence.

Comment 19: Your conclusion should be more explicit, as this only affects the test tube device used in this method and might be different for other devices.

Response 19: We have rewritten the conclusion to better reflect our findings.    

Comment 20: The impact, and current discussions, for potentially collecting cells/plasma from P1 to R1 has clinical implications for the bioformulation of the PRP; comment on the density of other cells in this regard as the PRP biology changes with their presence, as compared to a red-ish free PRP specimen.

Response 20: We can add something about the interactions of RBCs and WBCs with
platelets in PRP, but it's really not in the scope of this work. We are really only concerned with the relative movements of the different cell types.

Other changes to mention:

Means and Medians were provided: Although relative effects tables were made on the basis of median data, practitioners will benefit from seeing means and medians, and tables 1, 2, and 3 had both means and medians included.

A correction was made in calculation of tables 1 and 2 which was reflected in relative value tables: in order to calculate the mean platelet count for the addition of layers P1 and R1, both of which are ½ mL, the total platelet count should have been divided by two, since each platelet count was listed not as a total platelet count, but rather as a platelet count per mL. This did not appreciably affect the relative value tables, and P1+ R1 aspiration still outperformed P1+ 1/2P2.

We identified another aberrancy of data, as reflected in our new figure 2.   This figure clearly shows changes over time in mean platelet count. An examination of changes in the P1 layer let us to find an aberrancy of data is several samples in the P1 layer at the 20-minute period that resulting in a higher-than-expected platelet count in P1 at 20 minutes. In addition, the increase in platelet count in R5 was apparent and our impression is that related to the variable volume in that layer. Here is the portion of the comment we placed in the manuscript related to these two observations.

“It is notable upon examination of this table that the platelet count in layer P1 rose from 10 to 20 minutes, after dropping from 5 to 10 minutes, and that the platelet count in Layer R5 doubled from 10 to 20 minutes. A careful examination of 20 minutes data revealed an apparent data aberration in the P1 layer samples at the 20 minutes period. Absent those unexplained deviations, the P1 platelet count would likely have continued to drop at 20 minutes, but further data collection would be required for confirmation. A possible reason that platelet number increases in R5 at 20 minutes (and 10 minutes) is that, at these long centrifugation times the volume of R5 is more variable. See Figure 5, which shows the RBC layers become more compressed with longer centrifugation times. The R5 is open-ended.; i.e, the volume is whatever is left after R1-R4 have been aspirated. Thus the volume (and therefore the number of platelets) depended more on the volume than on the platelet concentration. Several samples had zero volume in R5 after 20 minutes (and were excluded from the calculations) so the dataset is also smaller.”

The length of the paper was expanded to 4410 words as a result of the very helpful efforts of the reviewers bring up areas in which comments should be included.

Thanks again for your time and help.

Regards,

Ted Harrison

Round 2

Reviewer 2 Report

Comments and Suggestions for Authors

Dear authors,

Thank you for your thorough review and significant changes to the manuscript. It has greatly improved! Thank you